# A Stochastic Multi-Attribute Method for Measuring Sustainability Performance of a Supplier Based on a Triple Bottom Line Approach in a Dual Hesitant Fuzzy Linguistic Environment

**DOI:** 10.3390/ijerph17062138

**Published:** 2020-03-23

**Authors:** Guohua Qu, Rudan Xue, Tianjiao Li, Weihua Qu, Zeshui Xu

**Affiliations:** 1School of Management Science and Engineering, Shanxi University of Finance and Economics, Taiyuan 030006, China; qugh@sxu.edu.cn (G.Q.); sc_ltj@163.com (T.L.); 2Institute of Management and Decision, Shanxi University, Taiyuan 030006, China; 3School of Economics and Management, Shanxi University, Taiyuan 030006, China; 4Business School, Sichuan University, Chengdu 610064, China; xuzeshui@263.net

**Keywords:** dual hesitant fuzzy linguistic term set, sustainability, group satisfaction degree, green supply chain, stochastic multi-attribute method

## Abstract

China is a developing country and with the speeding up of its industrialization, the environmental problems are becoming more serious, environmental pollution is a major environmental health problem in China. In order to have a more effective management and control of the significant growth issues of environment pollution, green supply chain incentives have started, which is kind of market incentive aiming to moderate the adverse effects of environmental pollution. Proper green chain supply selection and evaluation of companies is becoming very essential in sustainable green supply chain management. Generally speaking, decision-makers (DMs) prefer to provide a set of feasible and quantitative information for making performance evaluation, which motivates us to propose a framework using dual hesitant fuzzy linguistic term set (DHFLTS) and hesitant fuzzy linguistic term set (HFLTS) to select green suppliers. In this paper, group satisfaction and the regret theory are adopted for elicitation of preference information. The DHFLTS and HFLTS provide qualitative preferences of the DMs as well as reflect their hesitancy, inconsistency, and vagueness. Further, two new group satisfaction degrees are defined called the group satisfaction of hesitant fuzzy linguistic term set and dual hesitant fuzzy linguistic term set. Some properties of group satisfaction with DHFLST and HFL are also discussed. Unknown attribute weights are obtained to construct a novel Lagrange function optimization model to maximize the group satisfaction degree, which is an extension of general group satisfaction degree. A novel methodological approach based on two group satisfaction degrees framework and regret theory is developed to rank and select green chain suppliers focusing on specific selection objectives. The proposed model and method of this paper allow the DM to execute different fuzzy scenarios by changing importance weights attached to the triple-bottom-line areas. In the final part, the advantage of the proposed group satisfaction degree under DHFL and HFL background over the existing group satisfaction degree using examples have been presented with different computational combinations.

## 1. Introduction

Supply chain management (SCM) and their successful development have been paid attention by many scholars and practitioners in recent years. In an emotionally and physically distributed, heterogeneous, loosely-coupled environment of modern enterprise, outsourcing business has growing rapidly, becoming an integral part of supply chain management. This indicates logistics management or supply chain solutions play a major role in ensuring the competitiveness of supply chains. Sustainability has become an area of concern for corporate environmental practice that integrate environmental quality and social equity in their developing and implementing strategy. It must be stressed sustainable development is not only the development, but also the sustainability. Sustainability is a key factory in supply chains, is essential to the green economy and social development [1]. To meet demands of the economic order, people have recently realized that profits and profitability were only one element in the long-term survival and development [2]. Globalization and outsourcing have offered a kind of brand-new means for companies to create vast the supply chain integration network of retailers, suppliers, distributors, logistics and transportation providers as they search for the efficiency and superior returns promised by supply chains. In the course of implementing the sustainable development of strategy of supply chain management, it is inevitable that sustainability issues will arise from any link or kink in these activities. Some studies showed that the interaction between sustainability and supply chains is important, both for a “license to operate” and keeping competitive advantage of companies [3,4].

Giannakis et al. proposed an operational and system perspective of supply chain sustainability, by thinking of it as a risk management according to its process. It elaborates the nature of sustainability-related information risks of supply chain caused by “the bullwhip effect”, distinguishes their indicators from typical supply chain operating risks and develops an analytical procedure for their management and decision analysis [5]. Wolf points out that the impact of business stakeholders upon organizations’ adoption of better environmental and social practices is well documented in the existing literature of all countries. There is little focus on the relationship between stakeholder pressure and sustainable supply chain management (SSCM) [6]. To overcome this restriction, Wolf further investigated the relationship of coordination and realization of enterprise, social and environment and sustainable supply chain management, stakeholder pressure and corporate sustainability performance [6]. In general, according to target environment and information, organizations consider criteria such as availability, quality, price, flexibility, etc. in the application of evaluating supplier performance. In a modern society, the key factor of sustainable development plays a vital role for the basis of the long term success of a supply chain and the purchasing process becomes more interconnected and complicated with becoming increasingly of concern environmental and social pressures [7]. Sustainable supply chain is the direction which in modern industry is moving and it is the effective way to fulfill sustainable development for implementing green supply chain management. Nowadays, many organizations have considered triple-bottom-line (profit, people and planet) approach, which considers the economic social and environmental domains, has become a well-recognised term [7,8]. Creating value at three levels of sustainability-the economic, social and environmental have seen recent increased inclusion of environmental considerations both in practice and tool development and application [8,9].

In the real world, we often encounter some uncertain situations which are imprecise resulting from characterizing the fuzziness just by a membership degree in which it is difficult to use crisp numbers. Because the description of things by crisp numbers is not comprehensive enough, so people usually choose to use fuzzy and uncertain descriptions to express the understanding of things.

In this paper, we present a stochastic multi-attribute decision making approach based on dual hesitant fuzzy linguistic variable and hesitant fuzzy linguistic variable for evaluating sustainability performance of a supplier for triple bottom line. Based on above hesitant fuzzy linguistic variable and dual hesitant fuzzy linguistic variable, two new group satisfaction degrees we define two new concepts called the group satisfaction of hesitant fuzzy linguistic term set and dual hesitant fuzzy linguistic term set under the DHFLTS and the HFLTS.

In many real decision problems, due to the increasing insufficiency and complexity in available information and the lack of knowledge for the decision making process, it may be difficult for a decision maker to express his opinion with crisp values. Based on this consideration, it is more beneficial for decision makers to depict their preferences and characteristics using fuzzy linguistic variables rather than crisp numbers. This is the reason why we have used fuzzy set theory in sustainability performance evaluation of green chain suppliers in this paper. For example, it is much easier to represent the sustainability performance of green chain suppliers as high, very high, low, very low, etc., than in numbers. The decision makers give linguistic ratings, hesitant fuzzy linguistic ratings and dual hesitant fuzzy linguistic ratings to the sustainable performance-related criteria and to the alternatives (suppliers) which are then combined through a novel Lagrange function optimization model to maximize the group satisfaction degree to generate an overall performance score for each alternative. The advantage of maximizing the group satisfaction degree is that it distinguishes between benefit satisfaction degree (the bigger the better) and bad satisfaction degree (the smaller the better) category attributes and selects solutions that are close to the higher levels of satisfaction degrees and far from the lower levels of satisfaction degrees. The alternative with the highest score is finally chosen and recommended for procurement.

Application of hesitant fuzzy group satisfaction degrees and dual hesitant fuzzy group satisfaction for traditional supplier selection has been investigated by researchers in recent years. The research of linguistic variable information has become a hot research topic. Wan [10] proposed a mixed arithmetic aggregator based on two-tuple linguistic information. Zhu [11] et al. proposed a fast aggregated approach based on two-tuple linguistic information by converting the linguistic evaluated information into two-tuple linguistic comparison matrix. Tian et al. [12] studied the aggregation approach of three-point interval number complementary judgment matrix; Aiming at traditional linguistic group decision making, Baudry et al. [13] established an attribute-based optimization model and proposed a group decision-making method based on Monte Carlo empirical mode decomposition (EMD). Tseng [14] selected a more suitable alternative based on incomplete weight information and interactive conditions with respect to multiple green supply chain management (GSCM) criteria. The weight information about GSCM criteria and alternatives are expressed by a linguistic preferences that can be worked out by combining fuzzy set theory and an improved hierarchical model is introduced which provides a structured and logical method of synthesizing judgments that can be widely used by decision-maker for building an appropriate evaluation of suppliers. Zhang based on the multi-granularity linguistic judgment matrices, proposed a group decision-making approach incorporating with two optimization models which was proposed to aggregate these multi-format and multi-granularity linguistic judgments which was illustrated with a real-world case of horizontal directional drilling machine [15]. 

Recently, it has become increasingly common that assessments are provided by fuzzy numbers and intuitionistic fuzzy numbers. In 1965, Zadeh [16] proposed the theory of fuzzy sets to express the fuzziness in multi-attribute decision making problems. Since then, the theory of fuzzy sets has been developed rapidly. In 1986, Atanassov [17] proposed an intuitionistic fuzzy set, which takes into account the membership, non-membership and hesitation. Obviously, the intuitionistic fuzzy set is more flexible and practical than the fuzzy set. Atanassov [18] also proposed interval intuitionistic fuzzy sets. Since then, intuitionistic fuzzy sets have been further developed. 

In recent years, researchers have indicated that the application of fuzzy sustainability performance based on the triple bottom line approach used in the evaluation of supply chain flexibility is feasible. Lee et al. developed a model which was used to make evaluation according to the importance of the selected criteria and green supply chain management performance and the Delphi method was fully applied to differentiate the criteria which were evaluated traditional suppliers and green suppliers for the first time [19]. Anjali presented a fuzzy multi-criteria approach for evaluating environmental performance of suppliers. The proposed approach utilizes linguistic assessments used for rating and selecting the best green alternatives according to economic and social criteria. An integrated model was developed to obtain an overall performance score for each alternative combing fuzzy TOPSIS and these linguistic ratings [20]. Kannan applied a fuzzy TOPSIS approach for solving the green suppliers selecting with respect to the criteria of green supply chain management (GSCM) practices for a Brazilian electronics company. Kannan formulated a fuzzy TOPSIS-based supplier ranking model and applied an empirical analysis from a set of 12 available suppliers based on collected data [9]. In order to make internal improvements and selection of green suppliers clear, Liou et al. proposed an extended hybrid complex proportional assessment of alternatives with Grey relations (COPRAS-G) multi-attribute decision-making (MADM) model, where inter-dependent or interactive characteristics among various criteria and the vague information coming from decision-makers (DMs) are taken into account because of incomplete information, lack of knowledge and data or simply conflicts that are inherent between different groups within enterprises [9].

However, in real decision-making, the DMs often feel hesitant among multiple alternatives, and the traditional fuzzy set theory cannot fully reflect the decision making information. Therefore, Torra [21] proposed a hesitant fuzzy set (HFS), whose membership degree is composed of several sets of possible values, which can reflect the human’s hesitance more objectively. In fact, the HFS is a further extension of the theory of fuzzy sets. Xia and Xu [22,23] developed some operations and aggregation operators for hesitant fuzzy elements, and further studied the distance and similarity measure under hesitant fuzzy environment. Based on hesitant fuzzy sets, Chen et al. [24,25] introduced the concept of interval value and proposed interval-valued hesitant fuzzy sets. In fact, hesitant fuzzy sets only provide memberships, and it is difficult for the DMs to express their views accurately. On this basis, an extension of HFS has been presented. Zhu et al. [26] introduced the dual hesitant fuzzy set (DHFS), which permits the membership degree and non-membership degree having a set of possible values on the interval [0, 1]. Based on the existing research results, Zhu et al. [18] also proposed the operation rules, the accuracy function and the score function of dual hesitant fuzzy sets. Li and Su [27] proposed the definition of entropy and entropy formula of interval-valued dual hesitant fuzzy sets for DHFSs. Then, Ju et al. [28] defined interval-valued dual hesitant fuzzy sets (IVDHFS).

Due to the ambiguity and complex information in decision making problems, sometimes, the DMs often express the information by combining linguistic term set and fuzzy set. Under this condition, Wang et al. [29] combined intuitionistic fuzzy sets with linguistic information to define intuitionistic two-tuple linguistic information and intuitionistic fuzzy number. Liu et al. [30,31] also defined intuitionistic uncertain linguistic term set. Liu et al. [32] studied the fuzzy envelope based on the hesitant fuzzy linguistic term set. Lin et al. [33] presented some integration operators of hesitant linguistic fuzzy numbers. Liao et al. [34] proposed a similarity degree, consistency and inconsistency indices based on linguistic information to select an ERP system. Rodríguez et al. [34] introduced the concept of hesitant fuzzy linguistic term set. Since then, some decision making methods based on hesitant fuzzy linguistic sets have been proposed. Rodríguez et al. [35] also proposed a hesitant linguistic group decision making model based on the hesitant fuzzy linguistic term sets. Wang et al. [36] combined the outranking method with hesitant fuzzy linguistic term set to deal with the multi-criteria decision making problem. Ismat Beg et al. [37] proposed a modified version of fuzzy TOPSIS for hesitant fuzzy linguistic term sets. Liu et al. [38] proposed the hesitant intuitionistic fuzzy linguistic and some operators. Motivated by the idea of hesitant fuzzy linguistic variable, Yang and Ju [39] combined DHFS with linguistic term set to define the dual hesitant fuzzy linguistic set which contains a linguistic term, a set of membership degrees and non-membership degrees. 

As an important part of decision-making theory, behavioral decision-making receives more and more attention. Zhang et al. [40] paid attention to the application of prospect theory in risky multi-attribute decision-making. Wang et al. [41] combined prospect theory with TOPSIS, and proposed a decision making method. Liao et al. [42] studied the prospect theory under the hesitant fuzzy linguistic information. Prospect theory considers psychological factors such as risk preference, but it needs to find reference points artificially. The regret theory, proposed by Bell [43], Loomes and Sugden [44], is based on the regret psychology of the DMs and does not need to give reference information, which has attracted great attention of experts and scholars. Regret theory includes both the regret and joy of the DMs. In the real-life decision-making problems, the DMs should not only pay attention to the results of their choices, but also should not ignore the effects of other alternatives. If a DM finds that choosing other options will lead to better results, he may feel regretful. Therefore, regret theory has the characteristics of regret avoidance. Zhang and Fan [45] introduced regret theory into the risky multi-attribute decision-making problem, and established the matrix of regret value and rejoice value, and then ranked them according to the comprehensive perceived utility. Considering stochastic MCDM problems with interval probabilities, Zhou proposed a grey stochastic MCDM approach combined the VIKOR method and regret theory and Technique for Order Preference by Similarity to Ideal Solution (TOPSIS) [46]. 

Hesitant fuzzy linguistic term set includes a linguistic term and a membership set. Compared with the HFLS method, a dual hesitant fuzzy linguistic set provides non-membership additionally which can demonstrate problems more precisely. Dual hesitant fuzzy linguistic term set consists of three parts: linguistic evaluation phrases, membership and non-membership. It gives an assessment value both quantitatively and qualitatively. From the above analysis, we can see that there is no study on the regret theory under dual hesitant fuzzy linguistic term set environment and hesitant fuzzy linguistic term set environment. Therefore, this paper combines a dual hesitant fuzzy linguistic term set and a hesitant fuzzy linguistic term set with regret theory and group satisfaction, respectively. On this basis, we propose a group satisfaction based on dual hesitant fuzzy linguistic term set and hesitant fuzzy linguistic term set, and constructs regret and rejoice matrix. Finally, the degree of the alternative’s superiority and inferiority is obtained. The research on sustainability performance evaluation of green chain suppliers is however limited and needs more studies.

The hybrid methodology proposed in this paper uses a state of the art approach with a novel Lagrange function optimization model using a maximizing group satisfaction can start with a relatively small divergence of experts and the alternative ranking may arrive at a robust analysis of various green chain suppliers. The optimization model can prove practically valuable with its regret theory to consider the utility of the results of the current alternative and the regret-joy function. Theoretically speaking, this paper initiates a new direction of research by utilizing a series of comparison and stochastic tools for more effective green chain supplier selection in a continuous, complex and dynamic decision-making process. The approach, unlike other measures of optimization models, does not rely heavily on subjective inputs coming from green chain supply management professionals and managers; however scholars can very easily incorporate this additional information as prior incomplete weight information into the decision-making process. The optimization model may also utilize both qualitative (incomplete weight information) and quantitative (mathematical incomplete weight information) information for analysis.

The rest of the paper is organized as follows: In Section 2 and Section 3, we present preliminaries of hesitant fuzzy set and dual hesitant fuzzy set theory and sustainability supplier selection literature. In Section 4, we present a fuzzy multicriteria approach for evaluating sustainability performance of green chain suppliers. In Section 5, A new group satisfaction degree of DHFLTS and HFLTS are given. A decision-making method is illustrated in Section 6. A numerical application of the proposed approach is presented in Section 7. In Section 8, we conduct the comparison of the results. In Section 9. To improve their sustainability performance, we summarize the proposed model of the paper clearly, and an evaluation of the results from the illustrations helps to open the minds for managerial and research insights looking for directions for future research. 

## 2. Sustainability Supplier Selection Literature and Criteria

### 2.1. Green Supply Chain Management

Green supply chain management is the combination of natural environmental concerns and supply chain management [47]. The object of GSCM initiatives is to minimize the negative environmental impacts and waste of resources of the manufacturing and delivery of products and services [48]. At present, many scholars have made profound research on the meaning of green supply chain. Baumgartner [49] suggested that green supply chain management is the process of combining environmental issues into the corporate operations. GSCM focuses on how companies can use their processes and technology and how to integrate environmental issues to enhance their competitive advantage [50]. On this basis, Tseng et al. indicated that GSCM focuses not only on products but also on the materials resourcing [48]. According to current research results, Rao and Holt thought GSCM is the innovative management of supply chains in the areas of green housing, green manufacturing, green packaging and reverse logistics [51]. Inspired by Rao and Holt, Lin began to study the practice of green supply chain management from five aspects: green purchasing, green design, collaboration with suppliers and customers, and products recovery and reuse of used products [52]. The green supply management literature has found that supplier selection is a key part of green supply chain management [53].

### 2.2. Social Supply Chain Management

Corporate social responsibility (CSR) means that while pursuing maximum profits, enterprises should also take into account the responsibilities of the environment, employees, communities and other relevant stakeholders. CSR was proposed in the 1980s, with the continuous development of society, it has far-reaching impact and become one of the standards of enterprises. To accommodate more complex environmental sustainability, closed-loop supply chains (CLSCs) can be studied according to three activities, including recycling, remanufacturing, and reuse [54]. In the above-mentioned factors, reuse is considered to be one with the lowest cost [55]. To provide insight into the social sustainability, CSR activities are analyzed and explained by CLSC members [56]. 

The European Community Commission defined the CSR as voluntary, by integrating social and environmental issues. Carroll believed that corporate social responsibility should include economic responsibility, legal responsibility, moral responsibility and charitable responsibility [57]. With the development of society, corporate social responsibility is not only a constraint on the overall behavior of enterprises. Zhu et al. [58] proposed a novel methodological approach based on a Bayesian framework and Monte Carlo Markov Chain (MCMC) simulation to rank and select suppliers. Sustainable supply chain management (SSCM) obviously contributes to organizations’ performance [59]. 

### 2.3. Sustainable Supplier Selection Criteria Definition

One of the most important activities of supplier selection decisions is establishing the criteria system. Since 1960s, many researchers have focused on the establishment of these criteria [60]. However, it is not easy to obtain a criteria system that satisfies all of these suppliers. Dickson [55] indicated that the most important criteria are quality, delivery and performance history. Recently, Ho et al. proposed that the most criterion is quality. In addition to economic standards, environmental and social standards are also an indispensable part of supply chain management. In this study [61], we conclude some criteria that can be applied in the sustainable supplier selection in Table 1.

## 3. Preliminaries

### 3.1. Hesitant Fuzzy Linguistic Set

Some basic definitions related to hesitant fuzzy linguistic set are as follows:

**Definition** **1**[62]. *Let X be a fixed set, then a hesitant fuzzy linguistic set is defined as:*
(1)$$B=\left\{\left\langle x,s_{\theta \left(x\right)},h_{B}(x)\right\rangle |x\in X\right\}$$
*where*
$s_{\theta \left(x\right)}\in S=\left\{s_{0},s_{1},\cdots ,s_{l}\right\},$
$h_{B}(x)=\cup_{r_{B}\left(x\right)\in h_{B}\left(x\right)}\left\{r_{B}\left(x\right)\right\}$
*is a set of some values that lie in interval [0, 1], denoting the possible membership degrees of the element*
x∈X
*to the linguistic term*
$s_{\theta \left(x\right)}$*. For convenience,*
$b\left(x\right)=\left\langle s_{\theta \left(x\right)},h_{B}\left(x\right)\right\rangle$
*is called a hesitant fuzzy linguistic element (HFLE).*

**Definition** **2**[63]. *Let*
$b=\left\langle s_{\theta \left(b\right)},h\left(b\right)\right\rangle ,$
$b_{1}=\left\langle s_{\theta \left(b_{1}\right)},h\left(b_{1}\right)\right\rangle$
*and*
$b_{2}=\left\langle s_{\theta \left(b_{2}\right)},h\left(b_{2}\right)\right\rangle$
*be any three HFLEs, then the operational laws of HFLEs are defined as follows:*
$b^{\lambda }=\left\langle s_{\theta {\left(b\right)}^{\lambda }},\cup_{r\left(b\right)\in h\left(b\right)}\left\{r{\left(b\right)}^{\lambda }\right\}\right\rangle ;$$\lambda b=\left\langle s_{\lambda \times \theta \left(b\right)},\cup_{r\left(b\right)\in h\left(b\right)}\left\{1-{\left(1-r(b)\right)}^{\lambda }\right\}\right\rangle ;$$b_{1}⊗b_{2}=\langle s_{\theta \left(b_{1}\right)\times \theta \left(b_{2}\right)},\cup_{r\left(b_{1}\right)\in h\left(b_{1}\right),r\left(b_{2}\right)\in h\left(b_{2}\right)}\left\{r\left(b_{1}\right)r\left(b_{2}\right)\right\}\rangle ;$$b_{1}⊕b_{2}=\langle s_{\theta \left(b_{1}\right)+\theta \left(b_{2}\right)},\cup_{r\left(b_{1}\right)\in h\left(b_{1}\right),r\left(b_{2}\right)\in h\left(b_{2}\right)}\left\{r\left(b_{1}\right)+r\left(b_{2}\right)-r\left(b_{1}\right)r\left(b_{2}\right)\right\}\rangle .$

**Definition** **3**[38]. *For a HFLE, and the score function of b is defined as follows:*
(2)$$S\left(b\right)=\frac{s_{\theta \left(b\right)}}{l}\times \left(\frac{1}{\#h\left(b\right)}\sum_{r\in h\left(b\right)}r\right)$$
*where*
#h(b)
*is the number of elements in h(b),*
(l+1)
*is the cardinality of the linguistic term set*
S*. For two HFLEs b_1_ and b_2_, if*
$S\left(b_{1}\right)>S\left(b_{2}\right),$
*then*
$b_{1}>b_{2}$*; if*
$S\left(b_{1}\right)=S\left(b_{2}\right),$
*then*
$b_{1}=b_{2}.$

### 3.2. Dual Hesitant Fuzzy Linguistic Set

Based on dual hesitant fuzzy sets and linguistic evaluation sets, in order to facilitate the calculation, Yang et al. [63] proposed the dual hesitant fuzzy linguistic set and gave the corresponding operation rules, score values, etc., which are defined as follows:

**Definition** **4**[39]. *Let X be a fixed set, then a dual hesitant fuzzy linguistic set (DHFLTS)*
D
*on X is defined as:*
(3)$$D=\left\{\left\langle x,s_{\theta \left(x\right)},h(x),g(x)\right\rangle |x\in X\right\}$$
*where*
$s_{\theta \left(x\right)}\in S=\left\{s_{0},s_{1},\cdots ,s_{l}\right\},$
h(x)
*and*
g(x)
*are two sets of some values that lie in interval in [0, 1], denoting the possible membership degrees and non-membership degrees of the element*
x∈X
*to the linguistic term set*
$s_{\theta \left(x\right)}$*, respectively, with the conditions:*
$0\leq \gamma ,\eta \leq 1,0\leq \gamma^{+}+\eta^{+}\leq 1,$
*where*
γ∈h(x),
η∈g(x),
$\gamma^{+}=\cup_{\gamma \in h\left(x\right)}\max\left\{\gamma \right\},$
*and*
$\eta^{+}=\cup_{\eta \in g\left(x\right)}\max\left\{\eta \right\}$
*for all*
x∈X*. For convenience, the*
d(x)=
$\left\langle s_{\theta \left(x\right)},h\left(x\right),g\left(x\right)\right\rangle$
*is called a dual hesitant fuzzy linguistic element (DHFLE) denoted by*
$d=\left\langle s_{\theta },h,g\right\rangle .$

**Definition** **5**[64]. *Let*
$d_{1}\left(x\right)=\left\langle s_{\theta \left(d_{1}\right)},h_{1},g_{1}\right\rangle$
*and*
$d_{2}\left(x\right)=\left\langle s_{\theta \left(d_{2}\right)},h_{2},g_{2}\right\rangle$
*be two DHFLEs, then the operational laws are defined as:*
(1)$d_{1}⊕d_{2}=\langle s_{\theta \left(d_{1}\right)+\theta \left(d_{2}\right)},\cup_{\gamma_{1}\in h_{1},\gamma_{2}\in h_{2},\eta_{1}\in g_{1},\eta_{2}\in g_{2}}\left\{\gamma_{1}+\gamma_{2}-\gamma_{1}\gamma_{2}\right\},\left\{\eta_{1}\eta_{2}\right\}\rangle ;$(2)$d_{1}⊗d_{2}=\langle s_{\theta \left(d_{1}\right)\times \theta \left(d_{2}\right)},\cup_{\gamma_{1}\in h_{1},\gamma_{2}\in h_{2},\eta_{1}\in g_{1},\eta_{2}\in g_{2}}\left\{\gamma_{1}\gamma_{2}\right\},\left\{\eta_{1}+\eta_{2}-\eta_{1}\eta_{2}\right\}\rangle ;$(3)$\lambda d_{1}=\langle s_{\lambda \theta \left(d_{1}\right)},\cup_{\gamma_{1}\in h_{1},\eta_{1}\in g_{1}}\{\left\{1-{\left(1-\gamma_{1}\right)}^{\lambda }\right\},\left\{{\left(\eta_{1}\right)}^{\lambda }\right\}\}\rangle ,\lambda >0;$(4)$d_{1}^{\lambda }=\langle s_{\theta {\left(d_{1}\right)}^{\lambda }},\cup_{\gamma_{1}\in h_{1},\eta_{1}\in g_{1}}\{\left\{{\left(\gamma_{1}\right)}^{\lambda }\right\},\left\{1-{\left(1-\eta_{1}\right)}^{\lambda }\right\}\}\rangle ,\lambda >0;$

**Theorem** **1**[39]. *Let*
$d_{1}\left(x\right)=\left\langle s_{\theta \left(d_{1}\right)},h_{1},g_{1}\right\rangle$
*and*
$d_{2}\left(x\right)=\left\langle s_{\theta \left(d_{2}\right)},h_{2},g_{2}\right\rangle$
*be two DHFLEs, the calculation rules of DHFLEs are defined as follows:*
(1)$d_{1}⊕d_{2}=d_{2}⊕d_{1}$;(2)$d_{1}⊗d_{2}=d_{2}⊗d_{1}$;(3)$\lambda d_{1}⊕\lambda d_{2}=\lambda \left(d_{1}⊕d_{2}\right),\lambda >0$;(4)$d_{1}^{\lambda }⊗d_{2}^{\lambda }={\left(d_{2}⊗d_{1}\right)}^{\lambda },\lambda >0$.

**Definition** **6**[64]. *Let*
$d\left(x\right)=\left\langle s_{\theta },h,g\right\rangle$
*be a DHFLE, then the score function and the accuracy function of*
$d\left(x\right)=\left\langle s_{\theta },h,g\right\rangle$
*is defined as follows:*
(4)$$S\left(d\right)=\frac{\theta }{l}\times \left(\frac{1}{\#h}\sum_{\gamma \in h}\gamma -\frac{1}{\#g}\sum_{\eta \in g}\eta \right)$$
(5)$$P\left(d\right)=\frac{\theta }{l}\times \left(\frac{1}{\#h}\sum_{\gamma \in h}\gamma +\frac{1}{\#g}\sum_{\eta \in g}\eta \right)$$
*where*
#h
*and*
#g
*are the numbers of values in*
h
*and*
g*, respectively, (l+1) is the cardinality of*
$S=\left\{s_{0},s_{1},\cdots ,s_{l}\right\}.$

**Definition** **7**[39]. *Let*
$d_{1}\left(x\right)=\left\langle s_{\theta \left(d_{1}\right)},h_{1},g_{1}\right\rangle$
*and*
$d_{2}\left(x\right)=\left\langle s_{\theta \left(d_{2}\right)},h_{2},g_{2}\right\rangle$
*be any two DHFLEs, then:*
(1)*If*$S\left(d_{1}\right)>S\left(d_{2}\right),$*then*$d_{1}>d_{2}$*.*(2)*If*$S\left(d_{1}\right)=S\left(d_{2}\right),$*then:*(3)*If*$P\left(d_{1}\right)>P\left(d_{2}\right),$*then*$d_{1}>d_{2}$*,*(4)*If*$P\left(d_{1}\right)=P\left(d_{2}\right),$*then*$d_{1}=d_{2}$*.*

### 3.3. Group Satisfaction

In multi-attribute decision-making, different DMs will produce different preferences. In order to describe this situation, based on the score function and mean deviation function, Liu et al. [51] proposed a group satisfaction degree of hesitant fuzzy set, which is defined as follows:

**Definition** **8**[64]. *Let*
$h(x)={\left\{\gamma^{i}\right\}}_{i=1}^{l_{h}}$
*be a hesitant fuzzy element, and its group satisfaction degree is defined as follows:*
(6)$$\delta (h)=\frac{s(h)}{1+v(h)}=\frac{s(h)}{1+\frac{1}{l_{h}}\sum_{i=1}^{l_{h}}\left|\gamma^{i}-s(h)\right|}$$
*where*
$\gamma^{i}(i=1,2,\cdots ,l_{h})$
*represents the ith smallest value of the h(x). s(h) denotes the score function of the hesitant fuzzy element h(x), and v(h) represents the average deviation function of h(x), which demonstrates the degree of disagreement of the DMs.*

### 3.4. Regret and Rejoice in the Decision Making Process

Regret theory was proposed as a theory of choice under uncertainty. In recent years, regret theory has been widely used in multi-attribute decision making. As an important behavioural decision theory, regret theory was proposed independently by Bell, Loomes and Sugden [65], in 1982. The theory holds that, in the decision-making process, the final decision of a DM is not only affected by the results of the alternative he considers, but also affected by the results of other alternatives. If a DM finds that choosing other alternatives will lead to better outcomes, he may feel regretful; on the contrary, he will feel happy. As Zhang pointed out that regret theory is based on the assumption that people concern not only about what they receive but also about what they might have received [43,44]. Considering the actual decision-making process, the individual would compare the practical consequence with what the result would have been, and he would experience the emotions of regret and rejoicing as a consequence. Suppose that an individual had experienced regret when the consequences of the rejected action would have been better and rejoicing when the consequence of the rejected action would have been worse. Thus, to avoid post-decision regret from happening, the individual will take into account the anticipated regret and rejoicing in the decision-making process. 

Let M={1,2,…,m}, N={1,2,…,n}, T={1,2,…,t}. There is a set of alternatives that can be expressed as $Y=\left\{Y_{1},Y_{2},\ldots ,Y_{m}\right\}$, where *Y_i_* represents the *i*th alternative, $y_{i}$ represents the result of the alternative *Y_i_*, i∈M, $v(y_{i})$ represents the utility value of the alternative *Y_i_*. Let $H=\left\{H_{1},H_{2},\ldots ,H_{t}\right\}$ be a set of finite possibilities of observing status, $C=\left\{C_{1},C_{2},\ldots ,C_{j},\ldots ,C_{n}\right\}$ be a set of criteria, where *C_j_* represents the *j*th criterion, If $v\left(y_{i}\right)>v\left(y_{k}\right),\;k\in M$, then choosing the alternative $Y_{i}$ will feel happy; if $v\left(y_{i}\right)<v\left(y_{k}\right),i,k\in M,$ then choosing the alternative $Y_{i}$ will feel regretful.

According to regret theory, the perceived utility function of the DMs consists of two parts: the utility of the results of the current alternative and the regret-joy function. *y*_i_ and *y*_k_ denote the results of $Y_{i}$ and $Y_{k}$. The perceived utility of the alternative $Y_{i}$ is as follows: $U(y_{i},y_{k})=$
$v(y_{i})+Q(v(y_{i})-v(y_{k}))$, where *v*(*y_i_*) is the utility that the DM obtains from the alternative $Y_{i}$. *Q*(.) is a regret-rejoice function with respect to $v(y_{i})-v(y_{k})$. When $Q(v(y_{i})-v(y_{k}))≻0$, it implies that the DM will feel rejoice if they select the alternative *Y*_1_ instead of the alternative *Y*_2_. On the contrary, when $Q(v(y_{i})-v(y_{k}))≺0$, it implies that the DM will feel regret if they select the alternative *Y*_1_ rather than the alternative *Y*_2_. It is a monotone incremental concave function that satisfies $Q^{\prime }\left(\cdot \right)>0$, $Q^{\prime \prime }\left(\cdot \right)≺0$ and $Q^{\prime \prime }\left(\cdot \right)=0$ Regret-joy function is defined as follows:(7)Q(x)=1−exp(−βx)
where *Q* is exponential, skew-symmetric, continuous, and strictly increasing, β denotes the DM’s regret avoidance coefficient. β>0, and the bigger β is, the bigger regret avoidance is. β indicates the difference between the utility values of the two actions and then state of the world *H_t_* occurs. The attribute of *C_j_* Compared with *Y_k_*, the regret value of the alternative *Y_i_* is as follows:(8)$$Q_{ikj}^{t}=\{\begin{array}{cc} 1-\exp(-\beta (X_{ij}^{t})) & if\;X_{ij}^{t}=x_{ij}^{t}-x_{kj}^{t}≺0,\\ 0 & if\;X_{ij}^{t}=x_{ij}^{t}-x_{kj}^{t}≻0 \end{array}$$
where $x_{ij}^{t}$ is the value of *i*th alternative with respect to *j*th criterion and $Q_{ikj}^{t}$ is preference function is a non-decreasing function of the difference between $x_{ij}^{t}$ and $x_{kj}^{t}$, and then state of the world *H_t_* occurs. 

When the state of the world *H_t_* occurs. The attribute of *C_j_* compared with the alternative *Y_k_*, the rejoice value of the alternative *Y_i_* is as follows:(9)$$\Gamma_{ikj}^{t}=\{\begin{array}{cc} 1-\exp(-\beta (X_{ij}^{t})) & if\;X_{ij}^{t}=x_{ij}^{t}-x_{kj}^{t}\geq 0\\ 0 & if\;X_{ij}^{t}=x_{ij}^{t}-x_{kj}^{t}≺0 \end{array}$$

Obviously, according to Equations (8) and (9), $Q_{ikj}^{t}≺0$ if $x_{ij}^{t}≺x_{kj}^{t}$, and $Q_{ikj}^{t}=0$ otherwise; $\Gamma_{ikj}^{t}≻0$ if $x_{ij}^{t}≻x_{kj}^{t}$, and $\Gamma_{ikj}^{t}=0$ otherwise,.

Compared with the alternative *Y_k_*, the regret and rejoice value of the alternative *Y_i_* is as follows:(10)$$L_{ikj}^{t}=Q_{ikj}^{t}+\Gamma_{ikj}^{t}$$

## 4. Problem Description

Let $Y=\left\{Y_{1},Y_{2},\ldots ,Y_{m}\right\}$ be a set of *m* alternatives, $C=\left\{C_{1},C_{2},\ldots ,C_{j},\ldots ,C_{n}\right\}$ be a set of *n* attributes, $W=\left\{\omega_{1},\omega_{2},\ldots ,\omega_{n}\right\}$ be a set of parameter weights, where $\omega_{j}\in \left[0,1\right]$ with $\sum_{j=1}^{n}\omega_{j}=1$. Let $S=\left\{s_{0},s_{1},\ldots ,s_{l}\right\}$ be a finite linguistic term set. Furthermore, let $H=\left\{H_{1},H_{2},\ldots ,H_{t},\ldots ,H_{f}\right\}$ be the state of nature, $p=\left\{p_{1},p_{2},\ldots ,p_{t}\right\}$ with the conditions: $p_{t}\in \left[0,1\right]$ and *p*_t_ is probability of the state of nature such that $\sum_{t=1}^{f}p_{t}=1$. 

In this paper, the attribute values are expressed in the form of hesitant fuzzy linguistic term set and dual hesitant fuzzy linguistic term set, suppose that $D={\left(d_{ij}^{t}\right)}_{m\times n}$ is a dual hesitant fuzzy linguistic decision matrix, where $d_{ij}^{t}=\left\langle s_{\theta \left(d_{ij}^{t}\right)},h_{ij}^{t},g_{ij}^{t}\right\rangle \left(i=1,2,\ldots ,m;j=1,2,\ldots ,n;t=1,2,\ldots ,f\right)$ is in the form of DHFLEs given for the alternative set $Y=\left\{Y_{1},Y_{2},\ldots ,Y_{m}\right\}$, with $s_{\theta \left(d_{ij}^{t}\right)}\in S$. 

Similarly, the values assigned to alternatives are expressed by a hesitant fuzzy linguistic decision matrix denoted by $B={\left(b_{ij}^{t}\right)}_{m\times n}$, where $b_{ij}^{t}=\left\langle s_{\theta \left(b_{ij}^{t}\right)},h_{ij}^{t}\right\rangle \left(i=1,2,\cdots ,m;j=1,2,\cdots ,n;t=1,2,\cdots ,f\right)$ are HFLEs with $s_{\theta \left(d_{ij}^{t}\right)}\in S$.

## 5. A New Group Satisfaction Degree of DHFLTS and HFLTS

The group satisfaction Equation proposed by Liu et al. [66] not only uses the score function of hesitant fuzzy sets, but also introduces the average deviation function, which reduces the influence of subjective factors. Compared with hesitant fuzzy sets, hesitant fuzzy linguistic term set provide linguistic evaluation to describe the DM’s information which is more accurate. 

Dual hesitant fuzzy linguistic term set not only increase the linguistic evaluation phrases, but also provide non-memberships. In order to solve the corresponding decision-making problems, this paper proposes the group satisfaction of hesitant fuzzy linguistic term set and dual hesitant fuzzy linguistic term set.

**Definition** **9.**
*Let*
$d=\left\langle s_{\theta },h,g\right\rangle$
*be a DHFLE on*
x∈X
*, and its average deviation function is as follows:*
(11)$$\sigma \left(d\right)=\frac{1}{\#h}\sum_{\gamma \in h}\left|\frac{\theta }{l}\gamma -P\left(d\right)\right|+\frac{1}{\#g}\sum_{\eta \in g}\left|\frac{\theta }{l}\eta -P\left(d\right)\right|$$
*where*
#h
*and*
#g
*are the numbers of values in*
h
*and*
g
*, (l+1) is the cardinality of*
$S=\left\{s_{0},s_{1},\cdots ,s_{l}\right\}.$
P(d)
*is the accuracy function of*
$d=\left\langle s_{\theta },h,g\right\rangle$
*.*
σ(d)
*represents the degree of deviation to reflect the degree of divergence in decision-making.*


According to the accuracy function and the deviation functions of dual hesitant fuzzy linguistic sets, a new group satisfaction formula is defined in this paper:

**Definition** **10.**
*Let*
$d=\left\langle s_{\theta },h,g\right\rangle$
*be a DHFLE on*
x∈X
*, its group satisfaction degree is defined as follows:*
(12)$$z\left(d\right)=\frac{P\left(d\right)}{1+\sigma \left(d\right)}$$


Some properties of z(d) are as follows:

**Proposition** **1.**
0≤z(d)≤1.
*If*
$d=\left\langle s_{\theta },\left\{\gamma \right\},\left\{\emptyset \right\}\right\rangle$
*, then*
$z\left(d\right)=\frac{\theta }{l}\gamma .$

*If*
$d=\left\langle s_{\theta },\left\{\emptyset \right\},\left\{\eta \right\}\right\rangle$
*, then*
$z\left(d\right)=\frac{\theta }{l}\eta .$
$\frac{z\left(d\right)}{z\left(d^{c}\right)}=\frac{P\left(d\right)}{P\left(d^{c}\right)},d^{c}=\left\langle s_{\theta },g,h\right\rangle$.


**Proof.** $0\leq z\left(d\right)=\frac{P\left(d\right)}{1+\sigma \left(d\right)}=P\left(d\right)\leq 1$.If $d=\left\langle s_{\theta },\left\{\gamma \right\},\left\{\emptyset \right\}\right\rangle$, then $P\left(d\right)=\frac{\theta }{l}\gamma ,$
σ(d)=0,, $z\left(d\right)=\frac{\theta }{l}\gamma .$If $d=\left\langle s_{\theta },\left\{\emptyset \right\},\left\{\eta \right\}\right\rangle$, then $P\left(d\right)=\frac{\theta }{l}\gamma ,$
σ(d)=0,
$z\left(d\right)=\frac{\theta }{l}\eta .$$\sigma \left(d^{c}\right)=\frac{1}{\#h}\sum_{\gamma \in h}\left|\frac{\theta }{l}\gamma -P\left(d^{c}\right)\right|+\frac{1}{\#g}\sum_{\eta \in g}\left|\frac{\theta }{l}\eta -P\left(d^{c}\right)\right|=\frac{1}{\#h}\sum_{\gamma \in h}\left|\frac{\theta }{l}\gamma -P\left(d\right)\right|+\frac{1}{\#g}\sum_{\eta \in g}\left|\frac{\theta }{l}\eta -P\left(d\right)\right|=\sigma (d)$$\frac{z\left(d\right)}{z\left(d^{c}\right)}=\frac{\frac{P\left(d\right)}{1+\sigma \left(d\right)}}{\frac{P\left(d^{c}\right)}{1+\sigma \left(d^{c}\right)}}=\frac{P\left(d\right)}{P\left(d^{c}\right)}.$ ☐

**Definition** **11.**
*Let*
$b\left(x\right)=\left\langle s_{\theta \left(x\right)},h_{B}\left(x\right)\right\rangle$
*be a HFLE on*
x∈X
*, its deviation function is defined as follows:*
(13)$$\sigma \left(b\right)=\frac{1}{\#h\left(b\right)}\sum_{r\in h_{B}\left(x\right)}\left|\frac{\theta }{l}r-S\left(b\right)\right|$$
*where*
#h(b)
*is the number of values in*
$h_{B}\left(x\right)$
*, (l+1) is the cardinality of*
$S=\left\{s_{0},s_{1},\cdots ,s_{l}\right\}.$
S(b)
*is the score function of*
$b\left(x\right)=\left\langle s_{\theta \left(x\right)},h_{B}\left(x\right)\right\rangle$
*.*


**Definition** **12.**
*Let*
$b\left(x\right)=\left\langle s_{\theta \left(x\right)},h_{B}\left(x\right)\right\rangle$
*be a HFLE on*
x∈X
*, then the group satisfaction degree is defined as follows:*
(14)$$z\left(b\right)=\frac{S\left(b\right)}{1+\sigma \left(b\right)}$$


**Proposition** **2.**
0≤z(b)≤1.
*If*
h(b)={r}
*, then*
$z\left(b\right)=\frac{\theta }{l}$
*;*

$\frac{z\left(b\right)}{z\left(b^{c}\right)}=\frac{S\left(b\right)}{S\left(b^{c}\right)},b^{c}=\left\langle s_{\theta },\left\{1-r^{i}\right\}\right\rangle$
*.*



**Proof.** $0\leq z\left(b\right)=\frac{S\left(b\right)}{1+\sigma \left(b\right)}=S\left(b\right)\leq 1$If h(b)={r}, then $S\left(b\right)=\frac{\theta }{l}r,$
σ(b)=0,
$z\left(b\right)=\frac{\theta }{l}r.$$\sigma \left(b^{c}\right)=\frac{1}{\#h\left(b\right)}\sum_{r\in h\left(b\right)}\left|\frac{\theta }{l}(1-r)-S\left(b^{c}\right)\right|=\frac{1}{\#h\left(b\right)}\sum_{r\in h\left(b\right)}\left|\frac{\theta }{l}(1-r)-\frac{\theta }{l}\times \left(\frac{1}{\#h\left(b\right)}\sum_{r\left(b\right)\in h\left(b\right)}\left(1-r\right)\right)\right|=\sigma (b).$ ☐

## 6. Decision Making Method

### 6.1. Determination of Attribute Weights

The attribute weight model in hesitant fuzzy linguistic environment is similar to the model in dual hesitant fuzzy linguistic environment. Therefore, we use the dual hesitant fuzzy group satisfaction as an example to introduce the attribute weight model.

In real decision making, due to time pressure or lack of knowledge about the problem, the information of attribute weights is often incompletely known. Therefore, we propose some models to determine the attribute weight. Obviously, the higher the group satisfaction of attribute values, the smaller the divergence of experts and the better the alternative. In order to find the attribute weights on the net flow of a given alternative, following linear programming model is answered by maximizing the objective function of group satisfaction degree:
$$\{\begin{array}{l} \begin{array}{l} \max Z^{’}=\max f(w)=\left\{\sum_{t=1}^{f}\sum_{i=1}^{m}\sum_{j}^{n}\left[P\left(d_{ij}^{t}\right)/1+\sigma \left(d_{ij}^{t}\right)\right]\times w_{j}\right\}\\ \mathrm{subject}\;\mathrm{to}\;w\in W \end{array}\\ \sum_{j=1}^{n}w^{2}{}_{j}=1,0\leq w_{j}\leq 1\;\forall j \end{array}$$
where *w* is the vector of criteria weights and *W* is the feasible weight space defined by the partial information provided by the DM.

To construct a Lagrange function: $L(w,\lambda )=\left\{\sum_{t=1}^{f}\sum_{i=1}^{m}\sum_{j=1}^{n}\left[P\left(d_{ij}^{t}\right)/1+\sigma \left(d_{ij}^{t}\right)\right]\times w_{j}+\frac{\lambda }{2}(\sum_{j=1}^{n}w_{j}^{2}-1)\right\}$ Let: $\left\{ \begin{array}{l} \partial L(w,\lambda )/\partial w=\left\{\sum_{t=1}^{f}\sum_{i=1}^{m}z(d_{ij}^{t})\left[P\left(d_{ij}^{t}\right)/1+\sigma \left(d_{ij}^{t}\right)\right]+\lambda w_{j}=0\right\}\\ \partial L(w,\lambda )/\partial \lambda =\left\{\sum_{j=1}^{n}w_{j}^{2}-1=0\right\} \end{array}\right.$ Solving it, we get:(15)$$\omega_{j}=\sum_{t=1}^{f}\sum_{i=1}^{m}\left[P\left(d\right)/1+\sigma \left(d\right)\right]/\sum_{j=1}^{n}\sum_{t=1}^{f}\sum_{i=1}^{m}\left[P\left(d\right)/1+\sigma \left(d\right)\right](j=1,2,\ldots ,n)$$

This paper obtains the utility value v(d) according to Equation (12), i.e., the group satisfaction P(d)/1+σ(d). We establish the alternative comparison matrices for different attributes, $Q_{j}^{t}={\left[Q_{ikj}^{t}\right]}_{m\times m}$ and $\Gamma_{j}^{t}={\left[\Gamma_{ikj}^{t}\right]}_{m\times m}$, which denote the regret value and rejoice value for the attribute *C_j_* on the alternatives *Y_i_* and *Y_k_* respectively when the state *H_t_* occurs. Furthermore, by the integration of all the nature states, the utility that the DM obtained, given that the alternative *Y_i_* rather than *Y_k_* is selected with respect to the attribute *C_j_*, *Q_ikj_*, $\Gamma_{ijk}$ is given by:(16)$$Q_{ijk}=\sum_{t\in H_{t}}p_{t}Q_{ikj}^{t}(i=1,2,\ldots ,m;k=1,2,\ldots ,m,j=1,2\ldots n)$$
(17)$$\Gamma_{ijk}=\sum_{t=1}^{f}p_{t}\Gamma_{ikj}^{t}(i=1,2,\ldots ,m;k=1,2,\ldots ,m,j=1,2,\ldots ,n)$$

The regret value matrix $Q={\left(Q_{ikj}^{t}\right)}_{m\times m}$ and the rejoice value matrix $\Gamma ={\left(\Gamma_{ikj}^{t}\right)}_{m\times m}$ are as follows:
$$\begin{array}{c} \begin{array}{cccc} \;\;Y_{1} & Y_{2} & \cdots & Y_{m} \end{array}\\ Q_{ikj}^{t}=\begin{array}{c} Y_{1}\\ Y_{2}\\ \vdots \\ Y_{m} \end{array}\left(\begin{array}{cccc} 0 & Q_{12j}^{t} & \cdots & Q_{1mj}^{t}\\ Q_{21j}^{t} & 0 & \cdots & Q_{2mj}^{t}\\ \vdots & \vdots & \vdots & \vdots \\ Q_{m1j}^{t} & Q_{m2j}^{t} & \ldots & 0 \end{array}\right) \end{array},\;\begin{array}{c} \begin{array}{cccc} \;\;Y_{1} & Y_{2} & \cdots & Y_{m} \end{array}\\ \Gamma_{ikj}^{t}=\begin{array}{c} Y_{1}\\ Y_{2}\\ \vdots \\ Y_{m} \end{array}\left(\begin{array}{cccc} 0 & \Gamma_{12j}^{t} & \cdots & \Gamma_{1mj}^{t}\\ \Gamma_{21j}^{t} & 0 & \cdots & \Gamma_{2mj}^{t}\\ \vdots & \vdots & \vdots & \vdots \\ \Gamma_{m1}^{t}{}_{j} & \Gamma_{m2j}^{t} & \ldots & 0 \end{array}\right) \end{array}$$
where $Q_{ikj}^{t}$ and $\Gamma_{ikj}^{t}$ are the regret value matrix and the rejoice value matrix for the value of the *j*th criterion *C*_1_ respectively, then the state *H_t_* occurs.

Second, we standardize the regret and rejoice valued matrices:(18)$$\bar{Q}=\frac{Q_{ikj}}{T}$$
(19)$$\bar{\Gamma }=\frac{\Gamma_{ikj}}{T}$$
where $T=\max\left\{\underset{i,k\in M}{\max}\left\{\left|Q_{ikj}\right|\right\},\underset{i,k\in M}{\max}\left\{\left|\Gamma_{ikj}\right|\right\}\right\},j=1,2,\ldots ,n$.

According to the attribute weight, we construct the comprehensive regret-joy matrix. $Q(Y_{i})$ represents the DMs’ comprehensive regret values for the alternative *Y_i_*_,_
*G*(*Y_i_*) represents the comprehensive rejoice values for the alternative *Y_i_*. The equations are as follows:(20)$$Q(Y_{i})=\sum_{i=1}^{m}\sum_{j=1}^{n}\omega_{j}{\bar{Q}}_{ikj},i=1,2,\ldots ,m;k=1,2,\ldots ,m,j=1,2,\ldots ,n$$
(21)$$\Gamma (Y_{i})=\sum_{i=1}^{m}\sum_{j=1}^{n}\omega_{j}{\bar{\Gamma }}_{ikj},i=1,2,\ldots ,m;k=1,2,\ldots ,m,j=1,2,\ldots ,n$$

To calculate the ranking value of the alternative *Y_i_*:(22)$$L(Y_{i})=Q(Y_{i})+\Gamma (Y_{i})$$

The larger the value of $L(Y_{i})$, the better the alternative.

### 6.2. An Approach for Dual Hesitant Fuzzy Linguistic Stochastic Multi-Attribute Decision Making

For a dual hesitant fuzzy linguistic stochastic multi-attribute decision making problem, let $Y=\left\{Y_{1},Y_{2},\ldots ,Y_{m}\right\}$ be a set of alternatives, $C=\left\{C_{1},C_{2},\ldots ,C_{j},\ldots ,C_{n}\right\}$ be a set of attributes, $W=\left\{\omega_{1},\omega_{2},\ldots ,\omega_{n}\right\}$ be a set of parameter weights, and $S=\left\{s_{0},s_{1},\ldots ,s_{l}\right\}$ be a set of finite linguistic terms. In order to be able to select the best alternative or to rank the alternatives, we shall develop a practical approach for solving a dual hesitant fuzzy linguistic stochastic multi-attribute decision making (DHFLSMADM) problems, a DHFLSMADM method takes into account values of the alternatives with respect to the attributes and the importance degrees of the attributes, which is represented by the attribute weights. However, the DM may be willing or able to provide only incomplete information on parameters because of various reasons such as time pressure, lack of knowledge or data, intangible or non-monetary attributes, limited attention, information processing capabilities, etc. The schematic diagram of the proposed approach for (DHFLSMADM) is provided in Figure 1, which shows a data process diagram model of the proposed framework. 

The framework decision data process diagram model can be summarized in the following steps:

Case 1. If the assessment is expressed by hesitant fuzzy linguistic decision matrix $B={\left(b_{ij}^{t}\right)}_{m\times n}$, we can develop the following steps:

Step 1. Normalize the decision matrix $B={\left(b_{ij}^{t}\right)}_{m\times n}$. Benefit attributes remain unchanged and cost attributes are changed as follows: ${\tilde{b}}_{ij}=\left\langle s_{l-\theta \left({\tilde{b}}_{ij}\right)},{\tilde{h}}_{ij}\right\rangle$ where ${\tilde{h}}_{ij}=\cup_{r_{ij}\in h_{h}}\left\{1-r_{ij}\right\},$(*l+*1) is the cardinality of $S=\left\{s_{0},s_{1},\ldots ,s_{l}\right\}.$

Step 2. Calculate the weight. Calculate the weight of each attribute by Equation (15).

Step 3. Construct regret value matrix and rejoice value matrix. Calculate the regret value and rejoice value of alternative *Y_i_* with respect to *Y_k_.* Then we construct $Q_{j}={\left(Q_{ikj}\right)}_{m\times m}$ and $\Gamma_{j}={\left(\Gamma_{ikj}\right)}_{m\times m}$ by Equations (16) and (17), respectively.

Step 4. Normalize the regret value matrix and the rejoice value by Equations (18) and (19), we can obtain ${\bar{Q}}_{j}$ and ${\bar{\Gamma }}_{j}$.

Step 5. Compute the comprehensive regret value *Q*(*Y_i_*) and rejoice value Γ(*Y_i_*) by Equations (20) and (21).

Step 6. Obtain the ranking value $L(Y_{i})$ by Equation (22), and sort the solutions according to the descending order. The bigger $L(Y_{i})$ is, the better the alternative is.

Step 7. End.

However, in real decision making, due to information limitation, we often express the assessment by dual hesitant fuzzy linguistic term set denoted by $D={\left(d_{ij}^{t}\right)}_{m\times n}$.

Case 2. Similarly, if the assessment information is provided by dual hesitant fuzzy linguistic, then Step 1 is a little deferent, and other steps are the same. We can develop the following steps:

Step 1. Normalize the decision matrix $D={\left(d_{ij}^{t}\right)}_{m\times n}$. Benefit attributes remain unchanged and cost attributes are changed as follows: ${\tilde{d}}_{ij}=\left\langle s_{l-\theta \left({\tilde{d}}_{ij}\right)},{\tilde{h}}_{ij},{\tilde{g}}_{ij}\right\rangle$ where ${\tilde{h}}_{ij}=\cup_{\gamma_{ij}\in h_{h}}\left\{1-\gamma_{ij}\right\},{\tilde{g}}_{ij}=\cup_{\eta_{ij}\in g_{h}}\left\{1-\eta_{ij}\right\}$, (*l+*1) is the cardinality of $S=\left\{s_{0},s_{1},\ldots ,s_{l}\right\}.$

Step 2. Calculate the weight of each attribute by Equation (15).

Step 3. Construct the regret value matrix and rejoice value matrix. Then construct the ${Q^{\prime }}_{j}={\left({Q^{\prime }}_{ikj}\right)}_{m\times m}$ and ${\Gamma^{\prime }}_{j}={\left({\Gamma^{\prime }}_{ikj}\right)}_{m\times m}$.

Step 4. Normalize the regret value matrix and the rejoice value by equations (18) and (19)

Step 5. Compute the comprehensive regret value $\;Q^{\prime }(Y_{i})$ and the rejoice value  by Equations (20) and (21).

Step 6. Obtain the ranking value $L^{\prime }(Y_{i})$, and sort the solutions according to the descending order. The bigger $L^{\prime }(Y_{i})$ is, the better the alternative is.

Step 7. End.

This paper presents a new group satisfaction equation under the environment of hesitant fuzzy linguistic term sets and dual hesitant fuzzy linguistic term sets, and studies the stochastic multi-attribute problem based on regret theory. Firstly, the hesitant fuzzy linguistic term set and dual hesitant fuzzy linguistic term set are combined with regret theory respectively, which can more precisely describe the complex environment. Secondly, we combine the hesitant fuzzy linguistic term set and dual hesitant fuzzy linguistic term set with group satisfaction respectively. The result avoids subjectivity and is convenient to calculate.

## 7. Numerical Example Analyzing

### 7.1. Background Analysis

It is well known that “green” principles and strategies have become vital for companies due to public awareness of their environment impacts. A company’s environmental performance is not only related to the company’s inner environment efforts, but also it is affected by the supplier’s environmental performance and image. For industries, environmentally responsible manufacturing, return flows, and related processes require green supply chains (GSCs) and accompanying suppliers with environmental or green competence. In recent years, how to determine suitable and green suppliers in the supply chain has become a key strategic consideration. Kannan [67] proposed a framework using fuzzy TOPSIS to help green chain management. Büyüközkan [9] combined Decision Making Trial and Evaluation Laboratory model (DEMATEL), the Analytical Network Process (ANP), and (TOPSIS) in a fuzzy environment for green supplier evaluation in a specific company. Govindan [68] proposed a fuzzy multi-criteria approach for measuring sustainability performance of a supplier based on triple bottom line approach.

To the best of our knowledge, there is no literature about combining dual hesitant fuzzy and hesitant fuzzy group satisfaction degree proposed with regret theory to evaluate green suppliers in the supply chain. Now we consider a green supplier selection problem in which alternatives are the green supplier to be selected and criteria are those attributes under consideration adapted from Büyüközkan [7]. A high-technology manufacturing center in an automaker desires to select a suitable green material supplier to purchase the key components of new products. After preliminary screening, three candidates $Y_{i}\left(i=1,2,3\right)$ remain for further evaluation. In order to accurately judge the real situation of each suppliers, this high-technology manufacturing center select the most suitable green supplier based on three benefit criteria: (1) ISO 14000 ($C_{1}$); (2) Quality ($C_{2}$); (3) Health and safety ($C_{3}$). This hierarchical structure of this decision-making problem is shown in Figure 1. The three alternatives Y*_i_* (*i* = 1,2,3) are evaluated by dual hesitant linguistic information, and the linguistic term set is $S=\{s_{0}=extremelylow,s_{1}=very\;low,s_{2}=low,s_{3}=medium,s_{4}=high,s_{5}=vrey\;high,s_{6}=$
extremelyhigh}. There will be two natural states in the future, and the probability of occurrence of no natural state is: *p*_1_=0.6, *p*_2_=0.4. Three decision makers evaluate the candidates $Y_{i}\left(i=1,2,3\right)$ with respect to the criteria *C_j_* (*j*=1,2,3), and construct the following two dual hesitant fuzzy linguistic stochastic decision matrices (see Table 2 and Table 3): 

### 7.2. The Decision-Making Process Based on Dual Hesitant Fuzzy Linguistic Set

Step 1. Because the three attributes are all benefit-oriented attributes, there is no need for standardization.

Step 2. We utilize the decision information given in Table 2 and Table 3 to obtain the group satisfaction of each dual hesitant fuzzy linguistic element, and calculate the weight according to Equation (15), then we have:W=(0.31,0.35,0.34).

Step 3. Construct regret value matrix and rejoice value matrix, according to Equations (16) and (17). (a = 0.3) The matrices of alternatives are obtained as follows (see Table 4, Table 5, Table 6, Table 7, Table 8 and Table 9):

Step 4. Normalize the above matrices: (see Table 10, Table 11, Table 12, Table 13, Table 14 and Table 15):

Step 5. Compute the comprehensive regret value Q(Y_i_) and rejoice value Γ(Y_i_) by Equations (20) and (21).

$Q(Y_{1})=-0.371,$$Q(Y_{2})=-0.786,$$Q(Y_{3})=-0.761$. $\Gamma (Y_{1})=0.859,$$\Gamma (Y_{2})=0.495,$$\Gamma (Y_{3})=0.618$

Step 6. Obtain the ranking value of these three programs by Equation (22).
$$L\left(Y_{1}\right)=0.488\;L\left(Y_{2}\right)=-0.291\;L\left(Y_{3}\right)=-0.143$$

Then we rank the alternatives Y_i_ (i=1, 2, 3) according to the descending order of L(Y_i_). Therefore, we find that: $Y_{1}≻Y_{3}≻Y_{2}.$ where the symbol “≻” means “superior to”. So the alternative *Y*_1_ is superior to the alternative *Y*_3_ and the alternative *Y*_2_.

### 7.3. The Decision-Making Process Based on Hesitant Fuzzy Linguistic Set

If we ignore the non-membership, then the dual hesitant fuzzy linguistic element will reduce to the hesitant fuzzy linguistic element. Firstly, the dual hesitant fuzzy linguistic sets in Table 1 and Table 2 will reduce to hesitant fuzzy linguistic sets, as shown in Table 16 and Table 17.

Step 1. Because the three attributes are all benefit-oriented attributes, there is no need for standardization.

Step 2. We utilize the decision information given in Table 3 and Table 4 to obtain the group satisfaction of each dual hesitant fuzzy linguistic element. Then we calculate the weight, and the attribute weight can be obtained as follows: $W^{\prime }=\left(0.30,0.40,0.30\right)$

Step 3. Construct regret value matrix and rejoice value matrix (*a* = 0.3). The matrices of alternatives are obtained as follows (see Table 18, Table 19, Table 20, Table 21, Table 22 and Table 23):

Step 4. Normalize above matrices by Equations (18) and (19) (see Table 24, Table 25, Table 26, Table 27, Table 28 and Table 29):

Step 5. We can calculate the comprehensive regret values and rejoice values:

$Q^{\prime }\left(Y_{1}\right)=-0.797,$$Q^{\prime }\left(Y_{2}\right)=-0.577,$$Q^{\prime }\left(Y_{3}\right)=-0.234;$$\Gamma^{\prime }\left(Y_{1}\right)=0.446,$$\Gamma^{\prime }\left(Y_{2}\right)=1.052,$$\Gamma^{\prime }\left(Y_{3}\right)=0.822$. So the ranking values of the assessment values $b_{ij}^{t}$ are as follows: $L^{\prime }\left(Y_{1}\right)=-0.351,$
$L^{\prime }\left(Y_{2}\right)=0.475,$
$L^{\prime }\left(Y_{3}\right)=0.587$. Since $L^{\prime }\left(Y_{3}\right)>L^{\prime }\left(Y_{2}\right)>L^{\prime }\left(Y_{1}\right),$ then we can get ranking as $Y_{3}≻Y_{2}≻Y_{1}$. Therefore, the most feasible alternative is *Y*_3_.

## 8. Results Comparison

In order to verify the effectiveness of the proposed methods, we apply them to the dual hesitant fuzzy environment and hesitant fuzzy environment. If we do not consider the linguistic term set, then the dual hesitant fuzzy linguistic element will reduce to the dual hesitant fuzzy element. Thus, the approach proposed by [68] can be used to rank these alternatives. According to the dual hesitant fuzzy decision matrix, the attribute weight can be obtained as: *W^n^* = (0.33,0.33,0.34).

Secondly, the regret value matrix and rejoice value matrix are as follows (see Table 30, Table 31, Table 32, Table 33, Table 34 and Table 35):

Then, we can calculate the comprehensive regret values and rejoice values:$$Q^{\prime \prime }\left(Y_{1}\right)=-1.344,Q^{\prime \prime }\left(Y_{2}\right)=-0.241,Q^{\prime \prime }\left(Y_{3}\right)=-1.387$$
$$\Gamma^{\prime \prime }\left(Y_{1}\right)=0.423,\Gamma^{\prime \prime }\left(Y_{2}\right)=1.321,\Gamma^{\prime \prime }\left(Y_{3}\right)=0.832$$
so the ranking values of the assessment values are as follows:$$L^{\prime \prime }\left(Y_{1}\right)=-0.921,L^{\prime \prime }\left(Y_{2}\right)=1.080,L^{\prime \prime }\left(Y_{3}\right)=0.555$$
and thus, we can get ranking as $Y_{2}≻Y_{3}≻Y_{1}$. 

If we do not consider the linguistic term set and non-memberships, then the dual hesitant fuzzy linguistic element will reduce to the hesitant fuzzy element, so the method proposed by Liu et al. [69] can be used to solve the above numerical example. For the sake of simplicity, we use the table to represent the results, which is shown as Table 36.

From the two results obtained in the dual hesitant fuzzy linguistic environment and hesitant fuzzy linguistic environment, we can find that $Q^{\prime }(Y_{1})=-0.797<Q(Y_{1})=-0.371,$$\Gamma^{\prime }(Y_{1})=0.445<\Gamma (Y_{1})=0.859,$ which means that, compared with the hesitant fuzzy linguistic information, the alternative $Y_{1}$ faces less regret loss and more joy in a dual hesitant fuzzy linguistic environment. The orders of the alternative $Y_{2}$ and the alternative $Y_{3}$ are the same, but the optimal alternatives are different. The main reason for intensity the difference is non-membership considered in this paper. 

From the two results obtained in the hesitant fuzzy linguistic environment and hesitant fuzzy environment, it can be seen that the orders of the alternative $Y_{2}$ and the alternative $Y_{3}$ are different.

To sum up, we can draw the following conclusions by comparing the results: On the one hand, the dual hesitant fuzzy linguistic term set and hesitant fuzzy linguistic term set contain linguistic information, which can enrich the connotation of hesitant fuzzy theory. On the other hand, the regret theory considers the psychological characteristics of the DM based on the bounded rational hypothesis. If we only consider the DHFS or the HFS, the calculation of group satisfaction only reflects the degree of hesitation which lacks persuasion. Based on dual hesitant fuzzy linguistic sets and hesitant fuzzy linguistic sets, this paper proposes a dual hesitant fuzzy linguistic multi-attribute decision-making method and a hesitant fuzzy linguistic decision-making method, and takes into account objective decision information.

## 9. Conclusions

This paper establishes a relationship between regret and group satisfaction degree built in stochastic multi-attribute methods. We show that the complete ranking of the multi-attribute stochastic method of the dual hesitant fuzzy linguistic and hesitant fuzzy linguistic based on regret theory and group satisfaction enables the DM to select the alternative that maximizes total tempered rejoice. A sustainable green supply chain selection problem has shown that the proposed model is valid and robust. Compared to the considered method, the developed procedure for multi-attribute decision takes advantage of all dual hesitant fuzzy linguistic information stated by a possible linguistic variable including a set of membership degrees and a set of non-membership degrees, which will not cause no any loss of information in the process of aggregation, but also due to the consideration of characteristics of a possible linguistic variable which has a set of membership degrees and a set of non-membership degrees. Therefore, it is quite reasonable and useful for real-world applications. 

The benefit of implementing a sustainable green supply chain initiative which relies on the sustainable supply management is indispensable. Sustainable green supplier environmental, recycling economy, harmonious society and social collaboration can play a significant role in fulfilling the “triple bottom line” returns and providing forever and permanent power for the sustained development of society. This paper focuses mainly on the environmental ISO 14,000, social, and economic attributes for sustainable green chain supplier evaluation based on the triple bottom line concept, regret theory and group satisfaction degree. A comprehensive analysis of sustainable green chain supply selection to select suppliers uses various dimensions of sustainability’s triple-bottom-line approach—ISO 14,000 as environment criteria, quality as economy criteria, health and safety concerns as society criteria, and cultivating sense of social responsibility—effectively evaluated performance of suppliers to aid in sustainable green chain supplier selection should consider all three dimensions simultaneously. In this paper we have introduced a fuzzy stochastic multi-attribute approach for measuring sustainability performance green supplier selection decisions with consideration of sustainability criteria and a numerical application was provided to demonstrate the proposed framework. First, the attributes for evaluating sustainable green chain supply performance are discerned based on in the existing literature. Second, the experts provide dual hesitant fuzzy linguistic ratings to the criteria and the alternatives, and regret theory and group satisfaction degree are used to obtain the ratings and to generate an overall performance evaluation score. 

In this paper, with the respect to the multi-attribute decision making problems in which the attribute values are in the form of DHFLE and HFLE, this paper studies the MADM approach under the dual hesitant fuzzy linguistic environment and hesitant fuzzy linguistic environment. Firstly, the group satisfactions of DHFLE and HFLEs are defined. Then, a novel approach with dual hesitant linguistic information and hesitant fuzzy linguistic information is proposed based on regret theory, and some models for determining the attribute weights are constructed. Finally, a practical example is given to illustrate the application of the proposed method. The results show that the proposed method is feasible and practical, and enriches the dual hesitant fuzzy set and hesitant fuzzy set decision theory. More importantly, the combination of regret theory can accurately reflect the psychological behavior of DMs. The prominent characteristic is that it extends the traditional group satisfaction which is only used under hesitant fuzzy environment. 

As Goindan has pointed out, one of the limitations of the paper is that we have given a hypothetical illustrative example rather than providing a real world application [65]. Practical issues pertaining to the understandability, traceability, verifiability and accuracy of these proposed dual hesitant fuzzy linguistic and hesitant fuzzy linguistic decision procedure would need to be demonstrated for operational validity and feasibility of this multi-attribute decision making methodology. The availability of the great amount of information required for decision criteria and data needed for the application of the theoretical methodology is one of the limitations to this feasibility of operational management across the enterprise. With the development of society, sustainable green supply chain managers should be advised to pursue this type of data, not just in terms of the application of the method, but also in terms of operation management of their skills and organization in the future. In the face of all the challenges, such as time pressure, lack of knowledge of the relevant disciplines, etc., in the course of decision-making process, the information possessed by thje DM is asymmetric, so symmetric and asymmetric information symmetric will result in different results of decision making. In addition, an extension of group satisfaction degree, a dual hesitant fuzzy linguistic stochastic decision matrix technique, to a sustainable green chain environment is investigated, where the importance degrees of combinations or their ordered positions are neither globally considered nor overall focus on the correlations among combinations or their ordered positions. In the future research, we will focus on the important application of the proposed method in the design of sustainable green chain supply.

## Figures and Tables

**Figure 1 ijerph-17-02138-f001:**
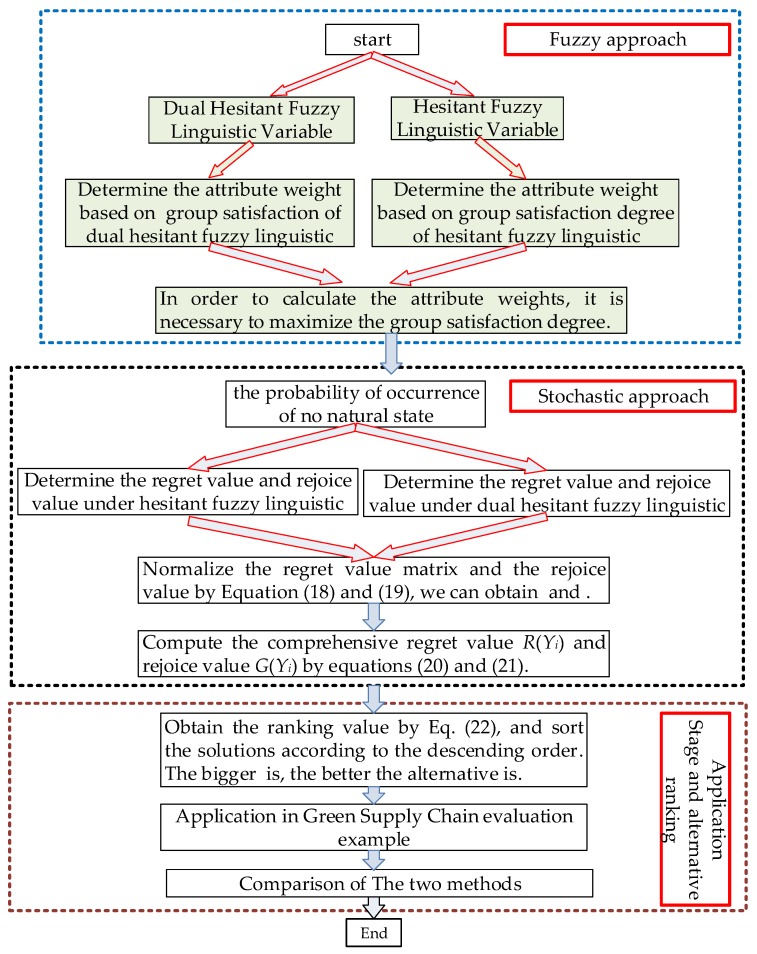
The schematic diagram of the proposed approach for DHFLSMADM.

**Table 1 ijerph-17-02138-t001:** Sustainable supplier selection criteria.

Three Dimensions	Criteria	Definition/Measures
Environmental	ISO 14000 (C_1_)	ISO14000 is environmental certifications which measures the enterprise’s environmental management system
Economic	Quality (C_2_)	Like quality of products
Social	Health and safety (C_3_)	Employee’s health, local communities’ health and safety incidents.

**Table 2 ijerph-17-02138-t002:** Dual hesitant fuzzy linguistic stochastic decision matrix (under state *H*_1_) *H*_1_(*p*_1_ = 0.6).

Candidates	C_1_	C_2_	C_3_
*Y* _1_	<s_3_, {0.4,0.6},{0.3,0.4}>	<s_4,_ {0.3,0.5},{0.2,0.3}>	<s_4,_ {0.2,0.4},{0.4,0.6}>
*Y* _2_	<s_2,_ {0.5,0.7},{0.2,0.3}>	<s_5,_ {0.4,0.6},{0.1,0.3}>	<s_3_, {0.4,0.5},{0.4,0.5}>
*Y* _3_	<s_4,_ {0.4,0.5},{0.2,0.4}>	<s_3,_ {0.5,0.7},{0.1,0.3}>	<s_3,_ {0.5,0.6},{0.2,0.4}>

**Table 3 ijerph-17-02138-t003:** Dual hesitant fuzzy linguistic stochastic decision matrix (under state H_2_) *H*_2_(*p*_2_ = 0.4)**.**

Candidates	C_1_	C_2_	C_3_
*Y* _1_	<s_4_, {0.2,0.6},{0.1,0.3}>	<s_4,_ {0.6,0.7},{0.2,0.3}>	<s_5_ {0.3,0.4},{0.3,0.5}>
*Y* _2_	<s_3_, {0.4,0.5},{0.3,0.4}>	<s_3,_ {0.6,0.8},{0.1,0.2}>	<s_4_ {0.6,0.7},{0.1,0.2}>
*Y* _3_	<s_4,_ {0.4,0.6},{0.1,0.3}>	<s_4,_ {0.4,0.6},{0.1,0.3}>	<s_3_ {0.4,0.6},{0.1,0.3}>

**Table 4 ijerph-17-02138-t004:** The regret matrix $Q_{1}$.

$Q_{1}$	*Y* _1_	*Y* _2_	*Y* _3_
*Y* _1_	0	0	−0.10
*Y* _2_	−0.14	0	−0.28
*Y* _3_	0	0	0

**Table 5 ijerph-17-02138-t005:** The regret matrix $Q_{2}$.

$Q_{2}$	*Y* _1_	*Y* _2_	*Y* _3_
*Y* _1_	0	−0.12	0
*Y* _2_	−0.01	0	−0.004
*Y* _3_	−0.01	−0.12	0

**Table 6 ijerph-17-02138-t006:** The regret matrix $Q_{3}$.

$Q_{3}$	*Y* _1_	*Y* _2_	*Y* _3_
*Y* _1_	0	0	0
*Y* _2_	−0.012	0	0
*Y* _3_	−0.026	−0.014	0

**Table 7 ijerph-17-02138-t007:** The rejoice matrix $\Gamma_{1}$.

$\Gamma_{1}$	*Y* _1_	*Y* _2_	*Y* _3_
*Y* _1_	0	0.014	0
*Y* _2_	0	0	0
*Y* _3_	0.12	0.028	0

**Table 8 ijerph-17-02138-t008:** The rejoice matrix $\Gamma_{2}$.

$\Gamma_{2}$	*Y* _1_	*Y* _2_	*Y* _3_
*Y* _1_	0	0.008	0.0104
*Y* _2_	0.012	0	0.014
*Y* _3_	0	0.004	0

**Table 9 ijerph-17-02138-t009:** The rejoice matrix $\Gamma_{3}$.

$\Gamma_{3}$	*Y* _1_	*Y* _2_	*Y* _3_
*Y* _1_	0	0.011	0.028
*Y* _2_	0	0	0.014
*Y* _3_	0	0	0

**Table 10 ijerph-17-02138-t010:** The normalized regret matrix ${\bar{Q}}_{1}$.

${\bar{Q}}_{1}$	*Y* _1_	*Y* _2_	*Y* _3_
*Y* _1_	0	0	−0.714
*Y* _2_	−0.5	0	−1
*Y* _3_	0	0	0

**Table 11 ijerph-17-02138-t011:** The normalized regret matrix ${\bar{Q}}_{2}$.

${\bar{Q}}_{2}$	*Y* _1_	*Y* _2_	*Y* _3_
*Y* _1_	0	−0.429	0
*Y* _2_	−0.357	0	−0.143
*Y* _3_	−0.357	−0.429	0

**Table 12 ijerph-17-02138-t012:** The normalized regret matrix ${\bar{Q}}_{3}$.

${\bar{Q}}_{3}$	*Y* _1_	*Y* _2_	*Y* _3_
*Y* _1_	0	0	0
*Y* _2_	−0.429	0	0
*Y* _3_	−0.929	-0.5	0

**Table 13 ijerph-17-02138-t013:** The normalized rejoice matrix ${\bar{\Gamma }}_{1}$.

${\bar{\Gamma }}_{1}$	*Y* _1_	*Y* _2_	*Y* _3_
*Y* _1_	0	0.5	0
*Y* _2_	0	0	0
*Y* _3_	0.429	1	0

**Table 14 ijerph-17-02138-t014:** The normalized rejoice matrix ${\bar{\Gamma }}_{2}$

${\bar{\Gamma }}_{2}$	*Y* _1_	*Y* _2_	*Y* _3_
*Y* _1_	0	0.286	0.371
*Y* _2_	0.429	0	0.5
*Y* _3_	0	0.143	0

**Table 15 ijerph-17-02138-t015:** The normalized rejoice matrix ${\bar{\Gamma }}_{3}$.

${\bar{\Gamma }}_{3}$	*Y* _1_	*Y* _2_	*Y* _3_
*Y* _1_	0	0.393	1
*Y* _2_	0	0	0.5
*Y* _3_	0	0	0

**Table 16 ijerph-17-02138-t016:** Hesitant fuzzy linguistic stochastic decision matrix (under state *H_1_*) *H*_1_(*P*_1_ = 0.6).

Candidates	C_1_	C_2_	C_3_
*Y* _1_	<s_3_, {0.4,0.6}>	<s_4,_ {0.3,0.5}>	<s_4,_ {0.2,0.4}>
*Y* _2_	<s_2,_ {0.5,0.7}>	<s_5,_ {0.4,0.6}>	<s_3,_ {0.4,0.5}>
*Y* _3_	<s_4,_ {0.4,0.5}>	<s_3_, {0.5,0.7}>	<s_5,_ {0.5,0.6}>

**Table 17 ijerph-17-02138-t017:** Hesitant fuzzy linguistic stochastic decision matrix (under state *H_2_*) *H*_2_(*P*_2_ = 0.4).

Candidates	C_1_	C_2_	C_3_
*Y* _1_	<s_4_, {0.2,0.6}>	<s_4_, {0.6,0.7}>	<s_5_, {0.3,0.4}>
*Y* _2_	<s_3_, {0.4,0.5}>	<s_3_, {0.6,0.8}>	<s_4_, {0.6,0.7}>
*Y* _3_	<s_4_, {0.4,0.6}>	<s_4_, {0.4,0.6}>	<s_3_, {0.4,0.6}>

**Table 18 ijerph-17-02138-t018:** The regret matrix ${Q^{\prime }}_{1}$.

${Q^{\prime }}_{1}$	*Y* _1_	*Y* _2_	*Y* _3_
*Y* _1_	0	0	−0.17
*Y* _2_	−0.11	0	−0.029
*Y* _3_	0	0	0

**Table 19 ijerph-17-02138-t019:** The regret matrix ${Q^{\prime }}_{2}$.

${Q^{\prime }}_{2}$	*Y* _1_	*Y* _2_	*Y* _3_
*Y* _1_	0	−0.022	−0.007
*Y* _2_	−0.11	0	0
*Y* _3_	−0.015	−0.022	0

**Table 20 ijerph-17-02138-t020:** The regret matrix ${Q^{\prime }}_{3}$.

${Q^{\prime }}_{3}$	*Y* _1_	*Y* _2_	*Y* _3_
*Y* _1_	0	−0.034	−0.028
*Y* _2_	0	0	−0.009
*Y* _3_	−0.005	−0.021	0

**Table 21 ijerph-17-02138-t021:** The rejoice matrix ${\Gamma^{\prime }}_{1}$.

${\Gamma^{\prime }}_{1}$	*Y* _1_	*Y* _2_	*Y* _3_
*Y* _1_	0	0.011	0
*Y* _2_	0	0	0
*Y* _3_	0.017	0.029	0

**Table 22 ijerph-17-02138-t022:** The rejoice matrix ${\Gamma^{\prime }}_{2}$.

${\Gamma^{\prime }}_{2}$	*Y* _1_	*Y* _2_	*Y* _3_
*Y* _1_	0	0.011	0.014
*Y* _2_	0.025	0	0.022
*Y* _3_	0.007	0	0

**Table 23 ijerph-17-02138-t023:** The rejoice matrix ${\Gamma^{\prime }}_{3}$.

${\Gamma^{\prime }}_{3}$	*Y* _1_	*Y* _2_	*Y* _3_
*Y* _1_	0	0	0.005
*Y* _2_	0.033	0	0.020
*Y* _3_	0.026	0.009	0

**Table 24 ijerph-17-02138-t024:** The normalized regret matrix ${{\bar{Q}}^{\prime }}_{1}$.

${{\bar{Q}}^{\prime }}_{1}$	*Y* _1_	*Y* _2_	*Y* _3_
*Y* _1_	0	0	−0.572
*Y* _2_	−0.33	0	−0.879
*Y* _3_	0	0	0

**Table 25 ijerph-17-02138-t025:** The normalized regret matrix ${{\bar{Q}}^{\prime }}_{2}$.

${{\bar{Q}}^{\prime }}_{2}$	*Y* _1_	*Y* _2_	*Y* _3_
*Y* _1_	0	−0.667	−0.218
*Y* _2_	−0.33	0	0
*Y* _3_	−0.455	−0.667	0

**Table 26 ijerph-17-02138-t026:** The normalized regret matrix ${{\bar{Q}}^{\prime }}_{3}$.

${{\bar{Q}}^{\prime }}_{3}$	*Y* _1_	*Y* _2_	*Y* _3_
*Y* _1_	0	−0.103	−0.848
*Y* _2_	0	0	−0.273
*Y* _3_	−0.145	−0.636	0

**Table 27 ijerph-17-02138-t027:** The normalized rejoice matrix ${{\bar{\Gamma }}^{\prime }}_{1}$.

${{\bar{\Gamma }}^{\prime }}_{1}$	*Y* _1_	*Y* _2_	*Y* _3_
*Y* _1_	0	0.330	0
*Y* _2_	0	0	0
*Y* _3_	0.515	0.879	0

**Table 28 ijerph-17-02138-t028:** The normalized rejoice matrix ${{\bar{\Gamma }}^{\prime }}_{2}$.

${{\bar{\Gamma }}^{\prime }}_{2}$	*Y* _1_	*Y* _2_	*Y* _3_
*Y* _1_	0	0.33	0.424
*Y* _2_	0.785	0	0.667
*Y* _3_	0.212	0	0

**Table 29 ijerph-17-02138-t029:** The normalized rejoice matrix ${{\bar{\Gamma }}^{\prime }}_{3}$.

${{\bar{\Gamma }}^{\prime }}_{3}$	*Y* _1_	*Y* _2_	*Y* _3_
*Y* _1_	0	0	0.152
*Y* _2_	1	0	0.606
*Y* _3_	0.789	0.273	0

**Table 30 ijerph-17-02138-t030:** The regret matrix ${Q^{\prime \prime }}_{1}$.

${Q^{\prime \prime }}_{1}$	*Y* _1_	*Y* _2_	*Y* _3_
*Y* _1_	0	−0.0072	−0.0036
*Y* _2_	0	0	0
*Y* _3_	−0.0054	−0.009	0

**Table 31 ijerph-17-02138-t031:** The regret matrix ${Q^{\prime \prime }}_{2}$.

${Q^{\prime \prime }}_{2}$	*Y* _1_	*Y* _2_	*Y* _3_
*Y* _1_	0	−0.0036	−0.009
*Y* _2_	−0.0012	0	−0.0054
*Y* _3_	−0.0072	−0.006	0

**Table 32 ijerph-17-02138-t032:** The regret matrix ${Q^{\prime \prime }}_{3}$.

${Q^{\prime \prime }}_{3}$	*Y* _1_	*Y* _2_	*Y* _3_
*Y* _1_	0	−0.0066	−0.036
*Y* _2_	0	0	0
*Y* _3_	−0.0024	−0.0054	0

**Table 33 ijerph-17-02138-t033:** The regret matrix ${\Gamma^{\prime \prime }}_{1}$.

${\Gamma^{\prime \prime }}_{1}$	*Y* _1_	*Y* _2_	*Y* _3_
*Y* _1_	0	0	0
*Y* _2_	0.0072	0	0.0036
*Y* _3_	0.0036	0	0

**Table 34 ijerph-17-02138-t034:** The regret matrix ${\Gamma^{\prime \prime }}_{2}$.

${\Gamma^{\prime \prime }}_{2}$	*Y* _1_	*Y* _2_	*Y* _3_
*Y* _1_	0	0.0012	0.072
*Y* _2_	0.0072	0	0.006
*Y* _3_	0.009	0.0054	0

**Table 35 ijerph-17-02138-t035:** The regret matrix ${\Gamma^{\prime \prime }}_{3}$.

${\Gamma^{\prime \prime }}_{3}$	*Y* _1_	*Y* _2_	*Y* _3_
*Y* _1_	0	0	0.0024
*Y* _2_	0.0066	0	0.0054
*Y* _3_	0.0036	0	0

**Table 36 ijerph-17-02138-t036:** Comparison of the results of the four decision making methods.

Alternatives	Dual Hesitant Fuzzy Element [65]		Hesitant Fuzzy Element(HFE) [69]		Dual Hesitant Fuzzy Linguistic Element(DHFLE)		Hesitant Fuzzy Linguistic Element(HFLE)	
	Regret Value	Rejoice Value	Regret Value	Rejoice Value	Regret Value	Rejoice Value	Regret Value	Rejoice Value
*Y* _1_	−1.344	0.423	−1.129	0.132	−0.371	0.859	−0.797	0.446
*Y* _2_	−0.241	1.321	−0.194	0.841	−0.786	0.495	−0.577	1.052
*Y* _3_	−1.387	0.832	−0.367	0.708	−0.761	0.618	−0.234	0.822
Ranking results	$Y_{2}≻Y_{3}≻Y_{1}.$		$Y_{2}≻Y_{3}≻Y_{1}.$		$Y_{1}≻Y_{3}≻Y_{2}.$		$Y_{3}≻Y_{2}≻Y_{1}.$	

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
