# Peer review of "A Stochastic Multi-Attribute Method for Measuring Sustainability Performance of a Supplier Based on a Triple Bottom Line Approach in a Dual Hesitant Fuzzy Linguistic Environment"

_ijerph, 2020, doi:10.3390/ijerph17062138_

Round 1

Reviewer 1 Report

Thanks for the efforts to contribute to the literature. Overall, you address what you did in the research. But, I have the following suggestions:

  1. Introduction should address clearly what you did and what you find. The current version is not acceptable. 
  2. Abstract: it should be more concise, it is too long. 

Thanks

Reviewer 2 Report

Dear Author,

the proposed method is interesting and well described.

Combine knowledge of scientific literature with a fuzzy approach can be applied to other important sector. 

I recommend a review of English language and style

Reviewer 3 Report

The research article presents an interesting model for decision making and/or potential evaluation of decision made in the past. Although it directly refers to a green supply chain and sustainability the model would be the same for any multivariable decision-making process. Maybe the authors should more clearly underline the fact that as regards CSR/greening etc. there are situations where there are a less economy and countable indicators, but more values where the fuzzy linguistic environment comes handy. The paper has a good structure and is clearly presenting the model, the method and the rationale behind it. There is a good account of the state of the art regarding the development of the used model. What is a weak point of the presented work is a used simplified data with only three benefit criteria. The usage of real-life data with additional description of the decision process would make the used model more clear and convincing for potential practitioners. However, the authors are aware of this. Accept in present form.

Reviewer 4 Report

The paper is interesting. It could be improved with some minor revisions. In detail:

- In the abstract there is a typing error in line 23. Moreover, I suggest to use a less jargon language (i.e. Generally speaken).

-In the introduction section the authors must check line 58 (they must eliminate T.) and line 61 (the number reference for Wolf).

- In the 2.2 paragraph the authors need to better explain the concept and the history of CSR. Researchers started to write and talk about it before the date indicated.

- In the paragraph 3.4 they must check the language (i.e. Suppose in line 334).

- In the paragraph 8 there are some errors (line 650 and line 658).

- The conclusions are clear.

If possible, I suggest to add more recent references (there are no references for 2019).
